# Novel Vaccine Strategies and Factors to Consider in Addressing Health Disparities of HPV Infection and Cervical Cancer Development among Native American Women

**DOI:** 10.3390/medsci10030052

**Published:** 2022-09-13

**Authors:** Crystal G. Morales, Nicole R. Jimenez, Melissa M. Herbst-Kralovetz, Naomi R. Lee

**Affiliations:** 1Department of Biology, Northern Arizona University, Flagstaff, AZ 86011, USA; 2Department of Obstetrics and Gynecology, College of Medicine, University of Arizona, Phoenix, AZ 85004, USA; 3Department of Basic Medical Sciences, College of Medicine, University of Arizona, Phoenix, AZ 85004, USA; 4Department of Chemistry and Biochemistry, Northern Arizona University, Flagstaff, AZ 86011, USA

**Keywords:** gynecology, vaginal microbiome, self-assembly, L2 capsid protein

## Abstract

Cervical cancer is the 4th most common type of cancer in women world-wide. Many factors play a role in cervical cancer development/progression that include genetics, social behaviors, social determinants of health, and even the microbiome. The prevalence of HPV infections and cervical cancer is high and often understudied among Native American communities. While effective HPV vaccines exist, less than 60% of 13- to 17-year-olds in the general population are up to date on their HPV vaccination as of 2020. Vaccination rates are higher among Native American adolescents, approximately 85% for females and 60% for males in the same age group. Unfortunately, the burden of cervical cancer remains high in many Native American populations. In this paper, we will discuss HPV infection, vaccination and the cervicovaginal microbiome with a Native American perspective. We will also provide insight into new strategies for developing novel methods and therapeutics to prevent HPV infections and limit HPV persistence and progression to cervical cancer in all populations.

## 1. Introduction

Cervical cancer is commonly diagnosed across the world, causing 342,000 deaths in 2020 [1,2]. Cervical cancer is a malignancy that affects the epithelial cells of the cervix, between the vagina and uterus. The main cause of cervical cancer is infection with human papillomavirus (HPV), a sexually transmitted infection with a high communicability [3,4]. In fact, HPV accounts for more than 95% of all cervical cancer cases [2,5]. Prior to HPV vaccine introduction, there were 14 million new infections a year. Once HPV vaccinations became available, the prevalence of infection dropped by 86% in females ages 14–19 years old and approximately 70% in females ages 20–24 years old [3,6]. Death rates from cervical cancer dropped significantly, however, more needs to be accomplished to increase vaccine uptake [1,2,7,8]. Despite having effective HPV vaccines available, less than 60% of 13- to 17-year-olds in the general U.S. are up to date with their vaccinations [9]. According to the CDC, approximately 33,000 HPV-associated cancers between the years 2014–2018 could have been prevented with higher vaccination uptake [8]. While Native American youth have high vaccination compared to the general public, Native American women still bear a heavier burden of cervical cancer and deaths when compared to white, non-Hispanic women in the U.S. [10,11]. The factors that play a role in this disparity will be discussed in this review. In addition, we will discuss the accessibility and efficacy of HPV vaccines among Native American women. 

## 2. Human Papillomavirus Prevalence 

Human papillomavirus (HPV) is the most common sexually transmitted infection with 70 million cases appearing in the anogenital tract in the U.S. prior to vaccine introduction [3,12]. Infection with high-risk HPV types are the main cause of cervical cancer, a cancer that is the 4th most prevalent among women around the globe [1,2]. Worldwide, there were an estimated 604,000 new cases of cervical cancer in 2020, with 342,000 deaths [2]. Unfortunately, this cancer burden disproportionately affects low-income countries, with about 90% of deaths occurring in developing countries [1,3]. 

Over 200 distinct genotypes of HPV have been identified, with an estimated 40 genotypes being able to infect the human mucosal epithelium [12]. As the most common virus to infect the reproductive tract, most sexually active men and women will be infected at least once, if not multiple times, in their lives. However, HPV is cleared by 90% of the infected population [1,7]. Concern arises with persistent infections, which cause an increased risk of developing serious malignancy precursors by more than tenfold [4,7]. 

### 2.1. ‘Old’ Controversy—Disproportionate Impact of HPV on Native American Women

In the U.S., 45,000 new cases of HPV-associated cancers (vagina, vulva, anus, penile, and oropharynx) arise each year among both men and women [8]. While national surveys report that approximately a quarter of the general U.S. female population aged 14–29 years test positive for at least one high-risk HPV, there are no large population or broad studies on HPV prevalence among Native American populations [13]. Recent tribe-specific studies have been able to estimate HPV prevalence in certain Native American populations in the US. The largest study included 698 Native American women from a single tribe located in the Great Plains. Among these women, 34.8% tested positive for at least one high-risk HPV [14]. Most alarming was discovering that Native American women over 30 years of age had an elevated HPV prevalence relative to the U.S. prevalence rate [11,15]. The reasons for the high prevalence are uncertain however, it is important to note older women were not eligible for the vaccines at the time. This is concerning due to increased cervical cancer rates among Native American women in the region [14]. Furthermore, recent studies including Native American women exhibit there are differences in the predominant HPV genotypes across two separate Native American communities [10,14]. In these studies, the most prevalent genotype was HPV-51, a genotype not included in the Gardasil 9_®_ HPV vaccine [14]. Clinical trials for HPV vaccines did not include Native populations and this bias resulted in a gap of protection. Therefore, Native American women are not as protected against prevalent HPV types as their white, non-Hispanic counterparts. As such, despite having higher vaccination rates compared to white, non-Hispanic women, there is evidence to suggest the current HPV vaccines are not as effective within the Native American populations therefore resulting in a disproportionate rate of HPV infections [10].

### 2.2. Current Strategies—Cervical Cancer Screening and HPV Testing

Across the over 570 federally recognized tribes, Native American women have higher rates of cervical cancer mortality compared to white, non-Hispanic women [10,11]. Specifically, older Native Americans have higher incidence of cervical cancer mortality. Though the reasons behind the disparities are not certain, it is likely due to various risk factors highlighted in Figure 1 [10,15,16]. The lack of screening can be due to several societal and environmental factors such as access to care, travel/transportation issues on the reservations, and/or lack of knowledge [10,15,17,18,19,20,21]. Per CDC recommendation, women starting at the age of 21 years old should get Papanicolaou (Pap) smears every three years. Starting at age 30, women are suggested to complete co-testing with both a Pap smear and HPV testing every five years [22]. In one study, over 400 cases of women diagnosed with invasive cervical cancer were analyzed for their screening history [16]. More than half of the women who failed to get screened, during the ideal period for prevention, were either characterized by low-education or high-poverty [10,16]. However, in this study and among other large studies, Native American women were excluded or listed as “other.” Underrepresented, underserved, and understudied women present with cervical and other gynecologic cancers at later stages, therefore resulting in poorer outcomes with regards to morbidity and mortality [23,24,25,26,27].

## 3. Current Knowledge on the Relationship between HPV, Cervical Cancer, and the Cervicovaginal Microbiome

The cervicovaginal microbiome is made up of bacteria, fungi, viruses, archaea and sometimes protists. A prototypical healthy cervicovaginal microbiome is often dominated by one or more species within the genus *Lactobacillus*, and assist in regulating homeostasis of this body site [28,29]. However, some women can be dominated by other bacteria such as *Prevotella*, *Gardnerella*, *Streptococcus* and never present symptoms or disease, thus further research on all cervicovaginal profile types are needed. The cervicovaginal microbiome is unique in its ecological structure in that high microbial diversity is not associated with cervicovaginal health versus other body sites, such as the gut, where diversity is associated with health [30,31]. Vaginal lactobacilli have a variety of mechanisms of exclusion such as production of lactic acid, which functions as a layer of protection and creates a resistant environment that minimizes the risk of viral infection and gynecological diseases [32,33]. Of the vaginal lactobacilli species, *Lactobacillus iners* is most common followed by *Lactobacillus crispatus*, *Lactobacillus gasseri/Lactobacillus paragasseri*, and *Lactobacillus jensenii/Lactobacillus mulieris* [34,35,36]. *L. crispatus* is often associated with cervicovaginal health and can produce a multitude of metabolites that function to bolster host immune responses, host epithelial cell adhesion, and exclude other bacteria and pathogens [29,37,38]

Cervicovaginal profiles with high abundance of lactobacilli can prevent HPV infection by increasing acidity levels via production of lactic acid [32]. A cervicovaginal microbiome that is more diverse with lower abundance of lactobacilli can increase the risk of HPV acquisition, persistent infection, and progression of neoplasia. Neoplasia may also alter the microbiome further, which could lead to clinical sequelae [28,39,40]. Composition of cervicovaginal profiles differs greatly between HPV positive and HPV negative subjects. In fact, patients with persistent high risk HPV infections had higher prevalence of bacterial vaginosis (BV) compared to patients that cleared HPV infections [32,39]. Interestingly, Brotman et al. described increased HPV clearance from patients who carried a high abundance of *L. gasseri*, however this was a small study [41]. 

Brusselaers et al., performed a systematic review of 14 observational studies reporting on incident HPV, HPV persistence, and cervical disease in women with or without a dysbiotic cervicovaginal microbiome. This group found that numerous studies provided a causal link between a dysbiotic cervicovaginal microbiome in HPV acquisition, persistence of HPV, and progression to dysplasia and cervical cancer [42]. A year later, the same group performed a follow-up systematic review and meta-analysis that described specific microbiota changes and reported a higher odds ratio for HPV infection and high-risk HPV progression to dysplasia/cervical cancer [39]. In this report of 11 articles, the authors state that cervicovaginal microbiome profiles containing dysbiotic microbes or *Lactobacillus iners* had increased risk of HPV infection and/or progression to cervical cancer compared with cervicovaginal microbiome profiles that were dominated by *Lactobacillus crispatus* [39]. These studies are foundational but have limitations with regards to confounding factors within the cohort such as including younger age women, women with more sexual partners, and high-risk sexual behavior. In addition, it was not mentioned by the authors that the cohort demographics included predominantly non-Hispanic, white women from the United States and European countries.

Other reports also found that HPV clearance is delayed when patients were diagnosed with BV, a dysbiotic vaginal microbial condition [28,43]. Roughly 43% of women with persistent HPV infection had a decrease in lactobacilli and a dominance of BV-associated bacteria (BVAB) such as *Gardnerella, Prevotella, Atopobium* (reclassified as *Fannyhessea*), and *Megasphaera* species compared to 7.4% of women with cleared HPV infection [32]. The shift in dominant bacterial species may be due to antiviral immune responses from NK and epithelial cells that are able to produce antimicrobial peptides (AMPs) [44,45]. Although AMPs are mostly associated with bacterial inhibition, new studies suggest an ability to inhibit viral pathogens as well [46]. Low levels of lactobacilli and an abundance of diverse anaerobic bacterial species has also been linked to HPV-mediated progression and severity of cervical cancers [39,42,47,48,49,50]. Laniewski et al., uniquely stratified groups by HPV status, dysplasia and newly diagnosed invasive cervical carcinoma, and identified key BVAB that may be important in cancer progression and included analyses on a racial/ethnically diverse cohort of women [50]. Usyk et al., observed that dysbiotic microbiota and *Gardnerella*
*spp.*, key players in BV, were more frequently observed in patients that progressed to cervical intraepithelial neoplasia ((CIN) [51]. It is hypothesized that a dysbiotic vaginal microbiome may contribute to this progression of CIN by modulating the host immune response, causing DNA damage or directly disrupting the cell barrier for oncoviruses such as HPV to infect the host more readily [52].

It has also been proposed that bacterial contributions from non-*Lactobacillus* dominated or *L. iners* dominated vaginal microbiome profiles are associated with increased inflammatory cytokines, sialidase-mediated epithelial barrier disruption, and biofilm formation, which may promote persistent infection, progression and invasion of carcinogenic cells [53,54,55]. Although the linkage between vaginal bacteria and gynecologic cancer has been more recently appreciated [39,56,57,58] several genera have been identified as putative gynecologically-relevant oncogenic bacteria: *Fannyhessea, Sneathia*, and *Porphyromonas* [59]. In vitro and clinical microbiome studies have revealed that these anaerobic bacteria produce metabolites or immune mediators that contribute to many hallmarks of cancer such as epithelial barrier disruption, immune and metabolic dysregulation, angiogenesis, and alterations in cellular proliferation [49,60,61]. Inflammation has been a large factor in severity of disease, and it is known that concurrent sexually transmitted infections affect genital inflammation [62,63,64,65,66,67,68].

Early on, it was debated whether HPV was a driver or passenger in cancer development; it is now known that some serotypes of the virus are drivers [69,70,71,72]. This same passenger-driver model has been proposed for bacteria and their relationship to cancer, questioning whether some bacteria directly cause cancer or merely favor the oncogenic environment for growth [48,73,74,75,76]. Evidence has been provided for direct and indirect effects of microbial composition with regards to cervical cancer development. For example, HPV has mechanisms that modify immune responses and the mucosal environment which are factors that impact microbial composition; however, BVAB also have mechanisms that increase genital inflammation and may promote an environment that favors sexually transmitted infection acquisition, including HPV infection [77,78,79,80]. It is not well understood if HPV alone, the BVAB linked with HPV, or the combination of the two are what lead to progression and development of cervical cancer and these factors require, and are the subject of, ongoing investigation [77,81,82,83,84,85].

### 3.1. Elucidating the Cervicoaginal Microbiome in Native American Populations in Context to HPV Infection

The cervicovaginal microbiome can be impacted by many external factors such as hormonal changes as in pregnancy and menopause, behavioral practices such as smoking and douching, and external/xenobiotic factors such as antibiotic usage [40,86,87,88]. Other factors such as host genetics can also play a role in microbial composition and in a recent study, patients with mutations in the BRCA1 gene associated with vaginal profiles with less than 50% abundance of *L**actobacillus* species [81,89,90,91,92]. Although there has been links to race and/or ethnicity being associated with specific vaginal microbiome composition [50,93,94,95], it is not fully understood whether these are biologically relevant features or features that relate to socioeconomic status, hygiene practices, geographic location, and diet [96]. Even fewer studies mention the impact on structural racism and oppression of minoritized groups and how that may impact microbial composition [97]. Further studies that engage with Native Americans and Indigenous populations often over-generalize these groups and do not account for distinct cultural and geographic differences of these racial/ethnic groups [98]. Thus, it is important to not draw conclusions from one cohort or tribal community and generalize to all Native American communities.

To this day, microbiome studies that have included Native American populations are limited [47,99,100,101]. However, one report incorporates data on the vaginal microbiome and psychosocial stress in Native American women [102]. This study revealed that in a cohort of 70 Northwestern Plains Native American women vaginal profiles consisted of *L. crispatus* dominant (n = 7), *L. iners* dominant (n = 17) or diverse anaerobes (n = 46) such as *Gardnerella, Prevotella, Atopobium*, and *Sneathia.* They also described an increase in anaerobes and bacterial biogenic amines that were highly significantly associated with lifetime trauma, historic loss, and stress [102]. Overall, in order to better resolve the HPV and cervical cancer health disparities amongst Native American populations, further vaginal microbiome studies are required across tribal communities, as well as the incorporation of additional social, behavioral and societal factors that may impact these minority groups.

### 3.2. Vaccine Development and Modulation of the Microbiome

It is clear that an increase in diverse bacteria in the cervicovaginal environment has a plethora of outcomes from preterm birth to cancer. Therefore, solutions have been proposed to help maintain homeostasis at this site. One approach to re-establishing a lactobacilli dominant vaginal environment is through the delivery of probiotics and/or prebiotics as a low-cost strategy with minimal known side effects [103,104,105,106,107]. Verhoeven et al., observed vaginal lactobacilli supplementation contributed to clearance of HPV [106]. Other studies have suggested postbiotic products of lactobacilli having antiviral capabilities which require further mechanistic investigation [108,109]. Further investigation into vaginal lactobacilli derived metabolites such as lactate or peptides could be exploited as a strategy for reinstating homeostasis of the vaginal microbiota and potential clearance [110]. Last, a vaginal microbiota transplant has been suggested as a strategy for modulating the microbiota [48,111,112,113]. Assessing the vaginal microbiome and relevant risk factors can provide critical insight on preventative, predictive, and personalized healthcare for patients with HPV infection, cervical dysplasia, or cervical cancer to improve individual outcomes and increase healthcare for the society.

Based on the relationship between the cervicovaginal microbiome and HPV infection, it is important to understand what is known on the effects of microbiome and vaccine efficacy. However, most of the existing literature on this topic has been established on the gut microbiome and not the vaginal microbiome [114,115,116,117,118,119,120,121]. Probiotic supplementation or pretreatment prior to HPV vaccination has shown an increased vaccine efficacy in the gut for both human and animal models [116,122]. This begs the question whether similar probiotic supplementation or microbiome modulation could be important in vaccine efficacy against vaginal pathogens. Some literature has investigated whether the composition of the gut microbiome could be predictive of vaccine efficacy [123,124,125]. A recent study by Ravilla et al. revealed that cervicovaginal microbiome profiles with high abundance of *Prevotella, Caldithrix, and Nitrospira* were less likely to elicit a protective immune response post-HPV vaccination, however, sample size and bacterial classification methods were limitations to the study cohort [126]. Giraldo et al. investigated whether the HPV vaccine, Cervarix^®^, affected microbial composition and immune response following vaccination [127]. This study indicated that after 7 months, there were no significant changes in the microbial composition of the patients and there were decreases in both pro-inflammatory and anti-inflammatory immune markers [127]. Despite the findings from cervicovaginal microbiome and HPV vaccines, Native American women were not included in these studies.

With regards to cervicovaginal vaccine development for the general population, and diverse communities such as Native Americans, it is important to note the complex interplay of systemic and local mucosal immunological factors, cervicovaginal microbiome composition and hormonal status (age, treatment, or contraceptive usage) on vaccine efficacy [128]. Hormonal status can impact the immune response to mucosal pathogens [128]. For example, the day of the menstrual cycle at which the vaccination is administered has been shown to contribute to HPV vaccination efficacy and the magnitude of the immune response [129,130]. Equally important to immunogenicity is how the microbiome primes the immune system, once again highlighting the complexity of vaccine development considerations [126,127,131]. Additional studies including vaccination status, hormone status, menstrual cycle, immune markers, and microbiome composition are needed to provide insights into the underlying mechanism for protective immunity and vaccine efficacy.

## 4. Established or Current Strategy: Designing HPV Vaccines for the General Population

HPV vaccines on the market today protect against specific high-risk and low-risk genotypes that can infect the oropharyngeal, genital, and rectal mucosal lining. Low-risk HPVs can result in cutaneous warts, while high-risk types are more likely to cause cancer. There are 14 oncogenic HPV types and 95% of cervical cancer cases in the general population are contributed to by high-risk HPV types [7,12,132]. HPV-16 and -18 alone account for 70% of cervical cancer and precancerous cervical lesions. Low-risk HPV types, such as HPV-6 and HPV-11 cause roughly 90% of all anogenital wart cases [7]. Currently, the primary method to prevent cervical cancer is through HPV vaccination [2,7,133]. There are no FDA-approved therapeutics to treat existing HPV infection; all current vaccines are prophylactic, meaning the vaccines are administered prior to infection [3,7]. However, there are therapeutic options for treating cervical dysplasia and therapeutic HPV vaccines are currently being investigated [7]. The approved HPV vaccine Gardasil 9^®^ protects against 9 HPV types (6, 11, 16, 18, 31, 33, 45, 52, and 58) and is effective at preventing warts and cancerous lesions in men and women [7,134].

### 4.1. Current Vaccine Target: L1

Belonging to the Papillomaviridae family, human papillomaviruses are small, double-stranded DNA, non-enveloped viruses. HPV’s genomes encode oncoproteins as well as two viral capsid proteins L1 and L2 [3,4,7]. Oncoproteins are gene products that create a favorable environment for replication and cell transformation. Infection occurs when there is an abrasion or opening in the epithelial layers and enables access to mitotically active basal cells. Newly synthesized virions are deposited in the outermost epithelial layer before viral shedding [3,4,7]. Epithelial cells of the skin, mouth, and anogenital mucosa are all targets for HPV infection and therefore vulnerable locations for cancer to develop. The different genotypes of HPV are classified by the L1 capsid protein sequence and numbered in chronological order of discovery [4]. HPV was one of the first human viruses discovered to have carcinogenic effects when it was linked to cancers of the cervix after HPV DNA was discovered in cervical cancer cells in the early 1980’s [4,12]. Current vaccines are composed of recombinant L1 capsid proteins because of their ability to spontaneously assemble into virus-like particles (VLPs) that are similar to native HPV virions, but without the viral replication machinery. These HPV VLPs induce effective, long-lasting immune responses without the infectious components of HPV [3,7,134]. However, HPV-specific antibodies following vaccination with these VLPs are type-specific [135,136]. Therefore, the vaccines are specific to the genotypes included in the vaccine formulation (e.g., HPV-6, 11, 16, 18, 31, 33, 45, 52, and 58). These HPV-specific antibodies neutralize live HPV, thereby preventing epithelial cells from being infected [7].

Since the implementation of HPV vaccines, there has been a significant drop in cervical cancer deaths. In the U.S. alone, the recorded cervical cancer deaths in 2012 were less than half of the deaths in 1975 [3]. High-income countries such as the U.S. are able to implement programs that allow adolescents to be fully vaccinated against HPV and women to be regularly screened for HPV and cervical cancer. Identifying pre-cancerous lesions at early stages allows for treatment prior to the development of cancer. Unfortunately, low-income countries have a decreased likelihood of vaccination due to cost; a U.S. pediatric dose of Gardasil 9^®^ through a CDC contract costs $208.05 and up to $253.60 in the private sector as of 2022 [137]. While there have been program initiatives to lower the cost for low-income countries, preventative measures and treatment are still limited, resulting in higher cervical cancer burden in countries such as Malawi, Uganda, the United Republic of Tanzania, Zimbabwe, and Zambia [1,138,139,140].

Vaccines may be unattainable due to the storage requirements of VLPs. A constant need for refrigeration from production to transport to distribution is called a cold chain and makes large-scale deliveries of these VLP based vaccines to rural areas very difficult. Another limitation of the current vaccines is low cross-protection to different HPV genotypes. The remaining high-risk HPV genotypes that are not in the current vaccines still contribute to 30% of all cervical cancers [7]. HPV-51, for example, is the most prevalent HPV type found in two geographically separate Native American communities [10]. These reasons highlight the need to pursue cost-efficient, thermostable, and broadly protective HPV vaccines.

### 4.2. Potential Vaccine Target: L2

As mentioned earlier, the HPV genome encodes for two different capsid proteins, the major L1 protein and the minor L2 protein. HPV has an icosahedral virus capsid consisting of 72 major L1 protein pentamers and in the center of each pentamer is the minor L2 protein [7]. Until recently, the purpose of the L2 protein was unknown; new research shows how critical this protein is to HPV infection. After L1 binds to a cell membrane, the viral capsid undergoes conformational changes and exposes the L2 protein. A secondary conformational change then allows viral uptake into the target cell and the L2 protein mediates delivery of the viral genome to the nucleus for transcription [3,7]. Recent research has shown the potential for anti-L2 antibodies to be cross-reactive and possibly give protection to heterologous HPV types. Since natural immunity does not induce L2 antibodies, there has been no evolutionary pressure for variation in the L2 sequence [141]. However, the exposed L2 protein during viral entry still allows this target to be accessible to antibody binding and neutralization [135,142,143].

Certain patterns and motifs are conserved within the N-terminus of the L2 protein so exploiting these conserved regions of the minor L2 protein could be a feasible tactic to broaden HPV protection [141,144]. Distinct from the L1 protein, the L2 protein cannot self-assemble into VLP structures, nor do they have high immunogenicity. However, studies have found that L2 proteins may become more immunogenic alongside an adjuvant or platform and in turn could become a successful HPV vaccine with long lasting immunity [7]. There is evidence suggesting that using a platform with multivalent display allows the immunogenicity of the L2 peptide to increase, such as being displayed on a bacteriophage VLP [141,143,145]. The ability to broaden coverage with the L2 protein could potentially provide more protection against HPV to populations who are more at risk, such as Native Americans.

In practice, studies have provided evidence that peptide sequences representing the N-terminus of the HPV-16 L2 capsid protein on bacteriophage VLPs can give rise to antibodies for a wide range of HPV types [141,145,146,147,148,149]. However, despite causing some degree of an antibody response, the N-terminal domain alone was not a sufficient antigen to induce protective, high-affinity, and neutralizing antibodies against a variety of HPV types [143]. To work around this problem, an HPV L2 consensus sequence was constructed from twelve high-risk HPV types (16, 18, 31, 33, 35, 39, 45, 51, 52, 56, 58, 59) and three low-risk types (6, 53, 66) by aligning all the sequences and choosing the most conserved residues (aa65–85) [143]. Experiments showed that mice immunized with this consensus L2 sequence (GTGGRTGYVPLGTRPPTVVDV) on VLPs were able to create antibodies and neutralize a wide spectrum of HPVs, specifically HPV-5, 6, 16, 18, 31, 33, 35, 39, 45, 51, 53, and 58 [141,150].

In another study, the authors showed that despite inducing a lower antibody titer count than HPV-16 or HPV-18 L2 proteins, the antibodies from the L2 consensus sequence were able to neutralize HPV-16, 18, 31, 45, and 58 [143]. This outcome shows that the antibodies induced by the L2 consensus peptide are not type specific and can cross-neutralize a range of high-risk HPVs [142,144]. While targeting the HPV L2 protein with a consensus sequence could be the basis of next-generation HPV vaccines for broader protection [146], more studies are required; as of 2020, there are several L2 vaccines in the clinical trial stage but most are in Phase I [151]. More updates on these vaccines may be shared in upcoming years.

### 4.3. Vaccines with Thermostability

Today’s VLP-based vaccines can suffer from reduced potency if not constantly kept at 2–8 °C, depending heavily on cold chains to maintain vaccine potency [152,153]. Cold chain is the term for the continuous refrigeration needed from production to travel to administration, but stable and constant electricity is not a possibility for every community. Poorly maintained refrigeration equipment, shortage of cold chain capacity, power outages, or inappropriate handling of shipments are all reasons why vaccines fail to be maintained at appropriate and optimal temperatures. Thermal sensitivity causes significant loss of vaccine material every year as high temperatures can cause unfolding of protein antigens, dissociation of polysaccharides, as well as reduction in the viability of attenuated vaccines [154,155]. The thermostability of Gardasil 4^®^ was evaluated at varying temperatures (25, 37, and 42 °C), of which, the half-life decreased from 130 months to approximately three months [152]. When spray dried, Gardasil 9^®^ maintained protection in mice, indicating the vaccine may be stored at 42 °C for up to 3 months without losing efficacy [156]. Within healthcare facilities, it is common practice to discard vaccines that have been exposed to high temperatures to ensure patient health. This problem creates a need for thermostable vaccines to prevent waste and allow for populations to obtain necessary vaccinations. In addition to strict storage requirements, vaccine delivery methods have other limitations such as potential cytotoxic side effects and complicated filtering/purifying processes [157]. One strategy that has proven to be thermostable and safe is peptide fibrils as vaccine platforms. Peptide platforms can give rise to strong immune responses without supplemental adjuvants [154].

## 5. New Strategies for Developing HPV Vaccines

### Self-Assembling Peptides Explained

Self-assembling peptides have the potential for a wide variety of biomedical applications including vaccine development and therapeutics. Peptides are the precursor to proteins and are made from amino acids, which are the building blocks of any protein. Proteins are crucial for normal body functions and play an important role in the immune system as well as other systems [157]. A specific category of peptides that are being researched for biomedical applications are self-assembling peptides due to their great biocompatibility in vivo, wide variety of function, and their biodegradable nature [157,158]. Self-assembly can be described as a spontaneous process of creating well-ordered structures from molecular units driven by non-covalent interactions such as hydrogen bonds, hydrophobic and electrostatic interactions, and π-π stacking interactions [157,158,159,160]. Complex structures organized by spontaneous alignment can be described in nature presenting as α-helices and β-sheets [157]. Cell proteins such as actin, tubulin and laminin are one dimensional self-assemblies while collagen and elastin are also self-assembling nanofibers [159]

Amphipathic peptides that have a sequence of alternating hydrophilic and hydrophobic residues spontaneously assemble into β-sheet bilayers when in aqueous solution seen in Figure 2 [157,161,162]. This assembly is driven by the hydrophobic residues burying the side chains inside the bilayer as hydrophilic residues form the outer layer [160,161]. A variety of structures can be formed from self-assembling β-sheets such as tapes, ribbons, and fibrils depending on the density of sheets that pack together [160,162]. Amphipathic β-sheets alone have many purposes in drug delivery such as hydrogel scaffolds and nanofibers [160,161,163].

Peptides are promising vaccine platforms due to their thermostability and ability to be stored for extended periods of time when lyophilized, eliminating the cold chain need [154]. Previous studies indicated that heating lyophilized self-assembling peptides at 45 °C for one to five weeks were stable and showed indistinguishable morphology. Likewise, the immunogenicity in mouse models did not diminish, even after heating the peptides to 45 °C for six months [154]. In addition, self-assembling peptides can also withstand varying solvents and pH, in contrast to current HPV vaccines [154,164,165].

Dissimilar to VLP’s, peptide fibrils are synthetic and therefore do not require bacterial expression systems for production which decreases the contamination risk [166]. Additionally, there are well established methods of peptide synthesis that produce high yields and allow for specific modifications [157]. Functionality in peptides can also be modified by the addition of any necessary compounds such as enzymes, drugs, antigens, and fluorescent compounds. Some self-assembling peptides can be used as a prophylactic vaccine delivery system with β-sheets fibrils peptides showing ability to elicit a strong immune response [162]. Current studies are underway that aim to broaden immunity and increase thermostability by using synthetic self-assembling peptides and the previously reported HPV consensus sequence [146]. We hypothesize that β-sheet peptides synthesized alongside the HPV consensus strand will create a more stable vaccine that will cause an immune response and evoke broadly neutralizing antibodies against a wide variety of oncogenic HPV types such as HPV-51, the most prevalent type identified in two Native American communities that is not covered by the current vaccines [Figure 2].

## 6. Conclusions

HPV-mediated cervical cancer disproportionately impacts Native American communities due to a network of complex factors such as immunological, behavioral, social, clinical, and environmental. In addition, the cervicovaginal microbiome has been associated with HPV acquisition, progression to cancer, and potentially HPV vaccine efficacy despite racial/ethnic background. Although, studies on Native American women are limited and require additional investigation. Furthering the health disparities amongst this population are hurdles such as access to HPV vaccines, cold chain, vaccination rates, and types of HPV genotypes included in vaccines. Self-assembling peptides as HPV vaccines may be a promising alternative to existing approaches and overcome mentioned obstacles. Using self-assembling peptides as a platform for the L2 consensus strand can be a cost-effective approach to constructing second generation HPV vaccines that are more globally accessible. The advantages of using peptide-based vaccines for both the platform and the antigen include known safety, biocompatibility, biodegradability, thermostability, and ease of production and storage. Additionally, the multivalent display of the HPV antigen on self-assembling peptides has the potential to boost the immune response to the L2 consensus sequence, allowing for enhanced protection. Storage requirements make the current HPV vaccines inaccessible to remote communities. The impact of using peptide technology alongside a novel HPV peptide antigen is the creation of a thermostable vaccine capable of eliciting a broad range of neutralizing antibodies. Pursuing these studies will be the first to show the effectiveness of immunizing with the HPV L2 consensus peptide on self-assembling β-sheet fibrils. An accessible prophylactic vaccine that has the potential to protect against the majority of HPV types can reduce rates of cervical cancer and potentially decrease global mortality rates. These strategies could aid in reducing the cancer health disparities within Native American and other communities that lack adequate access to healthcare.

## Figures and Tables

**Figure 1 medsci-10-00052-f001:**
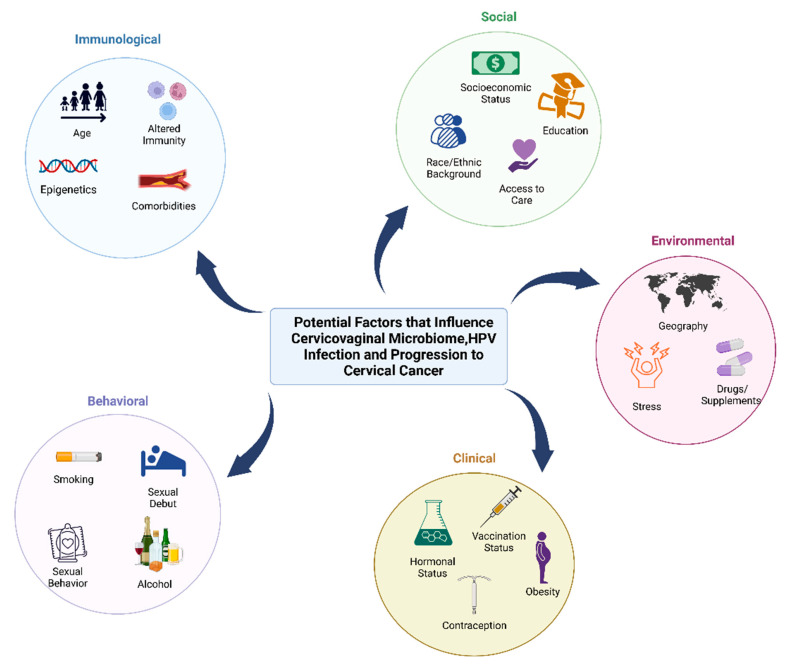
Behavioral, clinical, environmental, social, and immunological factors that may contribute to cervicovaginal microbiome, HPV infection, and progression to cancer. Factors that can impact the microbiome and host that increase the risk of HPV infection and cervical cancer can be influenced by behavioral factors (smoking, age of sexual debut, sexual activity such as use of lubricants and sex toys, contraception, feminine hygiene practices, and alcohol consumption), clinical factors (hormonal status, contraception, vaccination status, and hormonal dysregulation conditions such as obesity), environmental factors (geographic location, stress and trauma, and use of drugs/supplements or antibiotics), social factors (socioeconomic status, race or ethnic background, education level, and access to care), and immunological factors (age, epigenetics, altered immunity, and comorbidities such as cardiovascular disease).

**Figure 2 medsci-10-00052-f002:**
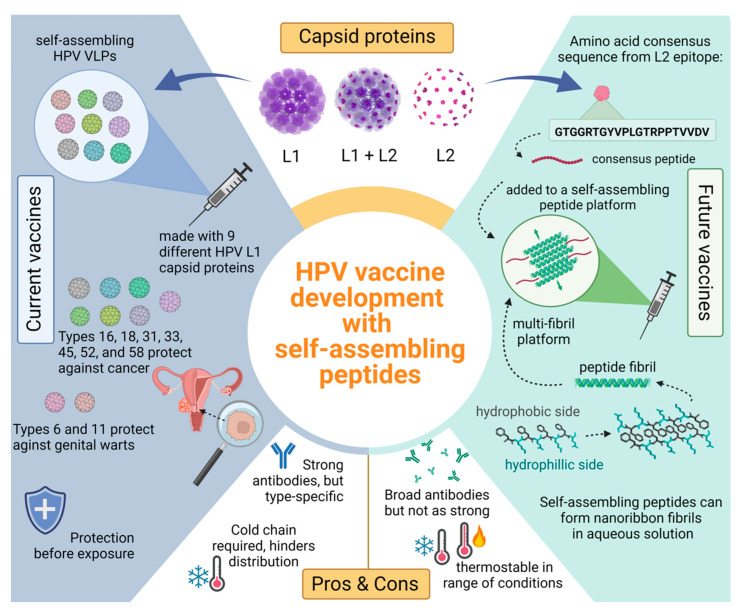
Comparing current L1 HPV vaccines with potential future L2 vaccines consisting of a self-assembling peptide platform. Top section shows an HPV capsid consisting of L1 (purple) and L2 (pink) proteins. The L1 proteins self-assemble into VLPs that are used in current HPV vaccines seen on the left blue side. Positive attributes to the current vaccines include strong and long-lasting antibodies against common HPV types; however, these antibodies are specific to the nine different L1 VLPs used. The green side showcases how a peptide platform vaccine would work with an L2 antigen. The bottom right shows a self-assembling peptide and how many of these peptides can self-assemble with anti-parallel stacking. These β-sheet fibrils present the L2 antigen in a multivalent display. Peptide platforms are highly thermostable compared to their VLP counterpart and the antibodies elicited may be broader, offering more coverage.

## Data Availability

Not applicable.

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
