# Peer review of "Novel Vaccine Strategies and Factors to Consider in Addressing Health Disparities of HPV Infection and Cervical Cancer Development among Native American Women"

_medsci, 2022, doi:10.3390/medsci10030052_

Round 1

Reviewer 1 Report

In the review titled “Novel vaccine strategies and factors to consider in addressing healthy disparities of HPV infection and cervical cancer development among Native American women”, the authors have provided a comprehensive survey of the factors that lead to higher prevalence of cervical cancer in Native American women and have discussed the possible strategies to develop novel HPV vaccines to help resolve these factors.

The authors have provided a detailed background of cervical cancer and HPV, along with a systematic and thorough overview of the current knowledge pertaining to the cervical microbiome, vaccine development strategies, and the drawbacks and strengths of the present methods used in vaccine development. Specially appreciated are the figures that provide a great graphic representation of the text in a concise and well-organized manner.

The reviews headings are well organized, and contain all the relevant information the reader may seek. For example, Section 4: Established or Current Strategy: Designing HPV Vaccines for the General Population, provides a detailed and very well written summary of the science behind vaccine design against HPV, and the shortcomings in the field. The following section 5: New Strategies for Developing HPV Vaccines, discusses the potential use of self-assembling peptides in the vaccine design that will help circumvent the shortcomings discussed in the previous section. 

Overall, the review is well-written, interesting to read and is of great value to the public health research community. I highly recommend accepting it in it's present form. 

Major concerns:

I believe there is a typo in the title (“healthy” disparities). Perhaps the authors meant “health” disparities.

Author Response

Dear Reviewer, 

We thank you for your time and comments. Below you will see our response to your comment. 
======================= 

Reviewers' Comments:  

In the review titled “Novel vaccine strategies and factors to consider in addressing healthy disparities of HPV infection and cervical cancer development among Native American women”, the authors have provided a comprehensive survey of the factors that lead to higher prevalence of cervical cancer in Native American women and have discussed the possible strategies to develop novel HPV vaccines to help resolve these factors.

The authors have provided a detailed background of cervical cancer and HPV, along with a systematic and thorough overview of the current knowledge pertaining to the cervical microbiome, vaccine development strategies, and the drawbacks and strengths of the present methods used in vaccine development. Specially appreciated are the figures that provide a great graphic representation of the text in a concise and well-organized manner.

The reviews headings are well organized, and contain all the relevant information the reader may seek. For example, Section 4: Established or Current Strategy: Designing HPV Vaccines for the General Population, provides a detailed and very well written summary of the science behind vaccine design against HPV, and the shortcomings in the field. The following section 5: New Strategies for Developing HPV Vaccines, discusses the potential use of self-assembling peptides in the vaccine design that will help circumvent the shortcomings discussed in the previous section. 

Overall, the review is well-written, interesting to read and is of great value to the public health research community. I highly recommend accepting it in it's present form. 

Minor concerns:

Comment 1: I believe there is a typo in the title (“healthy” disparities). Perhaps the authors meant “health” disparities.

Response: We thank the reviewer for catching the typo and revised the title on the edited draft to read “Novel vaccine strategies and factors to consider in addressing health disparities of HPV infection and cervical cancer development among Native American women”

Reviewer 2 Report

The review by Morales et al. “Novel vaccine strategies and factors to consider in addressing healthy disparities of HPV infection and cervical cancer development among Native American women” initially piqued my interest but left me with more questions than answers.  In order for me to recommend publication, the authors should carefully review each section and provide more clarity/information necessary for a review article on the topic. Sections in this manuscript have little to do with the title of their review, are not included in the abstract, and detract from the main points I believe the authors were trying to make.  It seems as if this manuscript was “pieced” together – sections seem disjointed and the information within a section doesn’t always match the title of the section.  Additionally, a very quick literature search revealed recent articles that are not included that specifically address points they are making in the manuscript, which makes me wonder what else is not included.  The following lists my major and minor concerns with the manuscript:

Major concerns:

1. My biggest concern is that I was expecting this manuscript to review novel vaccine strategies and disparities of HPV infection in Native American women.  While there are a few places these are discussed, a large portion of the manuscript is devoted to the gut microbiome, to a discussion of current (and proposed) HPV vaccines, and immune responses.  I had to search to find the few statements that are focused on the title of this review and about Native American women in particular.  Seems like there needs to be a revamp in the focus of this review.  Many statements throughout are vague and don’t summarize findings from the literature they cite.  I would have liked to see more summary of data in the manuscript instead of vague statements such as: “However, vaccination with L2 proteins alongside an adjuvant or on an immunogenic platform gives the possibility of long-lasting immunity [7].”  Or  “For example, when displayed on bacteriophage VLPs, L2 peptides can be highly immunogenic.”  What, specifically did these studies show?

2. The authors state that HPV infection rates are higher in Native American populations even though vaccination rates in this population are higher than for the non-Hispanic white population.  The authors conclude at the end of section 1 that they are going to discuss the accessibility of HPV vaccines in the Native American population, and later state that the higher rates of cancer may be due to “insufficient cervical cancer screenings and routine well woman exams”.  More explanation is needed – if women are being vaccinated at a higher rate but not being seen by their healthcare providers, that seems contradictory.  Also, a summary of data to support these claims would strengthen the manuscript.

3.  Figure 1 is referred to in several places, even as support for statements that are not included in the figure.  Where in Figure 1 is the information to support the sentence: “Even fewer studies mention the impact on structural racism and oppression of minoritized groups and how that may impact microbial composition [Figure 1].”  Where in Figure 1 is the information to support the sentence: “Further need for studies including vaccination status, hormone status, immune markers, and microbiome composition is needed to get a better picture of the underlying mechanism for adequate vaccine efficacy [Figure 1].” 

4.  The section “New Strategy – Elucidating the Vaginal Microbiome…” doesn’t discuss any new strategies that I could determine.

5. The statement “Therefore, novel solutions have been proposed beyond antibiotic usage to help maintain homeostasis at this [cervicovaginal environment] site” seems odd.  Antibiotic usage has been shown to disrupt normal flora, not maintain homeostasis.  One “novel” solution discussed in this review has been around for decades, and then the authors state that these have “proven antiviral activity” but do not discuss any of the studies supporting this claim.  They also do not review the status of using novel metabolites or peptides produced by lactobacilli, which would be interesting to the reader.

6.  In the section on “HPV, Cervical Cancer, and the Cervicovaginal Microbiome”, the authors write “The reason for the shift in dominant bacteria species may be due to antiviral immune responses [23,37].” What are these?  This would be interesting to read.  The next paragraph shifts focus away from HPV and cervical cancer to discuss GI bacteria and gastric cancer, which is outside the scope of this paper.  A casual mention of pathogens linked to cancer is OK and should be in an introductory paragraph, not within a section devoted to HPV and the cervicovaginal microbiome.  A quick search of the literature on the cervicovaginal microbiome revealed interesting papers not included that could have been added and strengthened this section – a few notable examples:  Usyk et al. 2020, Curty et al. 2020, Dai et al. 2021

7.  The section on “Vaccine development and modulation of the microbiome” focuses on the gut microbiome. How would this aid in HPV vaccine development?  The authors discuss “vaccines for treatment of rotavirus infection” (should be prevention, not treatment!) and discuss the link between that vaccine and the gut microbiome, but this vaccine is oral – what is the connection with the HPV vaccine, which is IM?  How does the cervical microbiome affect HPV vaccination efficacy?  This is not discussed.  Additionally, the authors point out “increased intestinal immunity” or “increased immune responses” but don’t summarize what they are. 

8.  In the section “Current Vaccine Targets: L1”, the authors provide a review of studies using the L2 capsid - should the section title be changed/modified to include L2?  More importantly, this section ends with the statement that targeting the L2 capsid protein may be the basis of next-generation vaccination.  However, the studies cited in this review are 5-10 years old (the authors don’t include newer studies, such as Yadav et al. 2020) nor discuss why these vaccines haven’t been developed.  Is there a reason why this information is a decade old yet L2 capsid proteins have not been incorporated into a vaccine?  What efficacy has been reported by these studies?  More discussion here would strengthen this manuscript.

9.  The authors make a point (in a couple of places) about the cold chain requirement as a factor affecting vaccination rates.  Earlier they state that vaccination rates are higher among Native Americans, which seems contradictory to their argument.  But in addition, a study published by Shank-Retzlaff et al. (2006) evaluated the thermostability of Gardasil and their data show prolonged stability if kept at least at room temperature.  This should be discussed.

10.  The conclusion makes statements that have not been supported or adequately addressed throughout the review.  For example: “HPV-mediated cervical cancer disproportionally impacts Native American communities due to a network of complex factors such as immunological, behavioral, social, clinical, and environmental.”  Where’s the information on social factors influencing rates of cervical cancer?  The authors state that one hurdle to address is access to HPV vaccines in the Native American population, yet they state that vaccination rates are higher in this population compared to the white population (and why just this comparison??). 

Minor concerns/edits:

1.  In the abstract, the authors state that cervical cancer is the fourth most common type of cancer.  I believe the authors meant to include “in women”.  While the WHO makes this statement according to 2018 data, the CDC lists “corpus and uterus” cancer as the fourth leading cause.  This should be clarified.

2. To be more precise, in the introduction (and other places – without line numbers it’s difficult to point out all instances) instead of saying “The main cause of cervical cancer is human papillomavirus,…” edit to read “The main cause of cervical cancer is infection with human papillomavirus,…”

3. In several places, the authors write “the HPV vaccine” when there are at least four to my knowledge.  Edit to using “HPV vaccines” or “HPV vaccination” (e.g., top of page 2: “Once HPV vaccination became available…” and “Despite having effective HPV vaccines available…”).  Also, Cervarix is misspelled (p. 7).

4.  Section 3, third line: change “species” to “genus”:  “…often dominated by one particular bacterial genus, Lactobacillus,…”.  Alternatively, authors could write “…often dominated by one or more species within the genus Lactobacillus…"

5. Figure 2: The legend mentions KFE8 beta-sheet fibrils but nowhere in the manuscript is a description of what KFE8 means.

6.  The authors should carefully proofread the manuscript prior to resubmitting. There are fragment sentences, grammatically incorrect sentences, and tense disagreements in several places.  Without line numbers, it’s hard to point them all out, but here are a few examples:

·         “Although there has been links to race and/or ethnicity having an impact on vaginal microbiome composition [41,78-80].”

·         “Such as HPV-51, the most prevalent HPV type found in two geographically separate Native American communities [10].”

·         “Deaths from cervical cancer rates dropped significantly however more needs to be done to increase vaccine uptake [1,2,7,8].”  (Deaths from a rate??)

·         “It is important to note that evidence has been provided for direct and indirect effects of microbial composition importance with regards to cervical cancer development.”

·         “…some serotypes of the virus are now established as a drivers…”

·         “…mutations in BRCA1 gene…” (needs a “the” before BRCA1)

Author Response

 Dear Reviewer, 

Thank you for taking the time to review our work. We have taken all comments and critiques very seriously. Below you will see our response and hopefully it will be received kindly.

======================= 

Reviewers' Comments:

The review by Morales et al. “Novel vaccine strategies and factors to consider in addressing healthy disparities of HPV infection and cervical cancer development among Native American women” initially piqued my interest but left me with more questions than answers. In order for me to recommend publication, the authors should carefully review each section and provide more clarity/information necessary for a review article on the topic. Sections in this manuscript have little to do with the title of their review, are not included in the abstract, and detract from the main points I believe the authors were trying to make. It seems as if this manuscript was “pieced” together – sections seem disjointed and the information within a section doesn’t always match the title of the section. Additionally, a very quick literature search revealed recent articles that are not included that specifically address points they are making in the manuscript, which makes me wonder what else is not included. The following lists my major and minor concerns with the manuscript:

Major concerns:

Comment 1. My biggest concern is that I was expecting this manuscript to review novel vaccine strategies and disparities of HPV infection in Native American women. While there are a few places these are discussed, a large portion of the manuscript is devoted to the gut microbiome, to a discussion of current (and proposed) HPV vaccines, and immune responses. I had to search to find the few statements that are focused on the title of this review and about Native American women in particular. Seems like there needs to be a revamp in the focus of this review. Many statements throughout are vague and don’t summarize findings from the literature they cite. I would have liked to see more summary of data in the manuscript instead of vague statements such as: “However, vaccination with L2 proteins alongside an adjuvant or on an immunogenic platform gives the possibility of long-lasting immunity [7].” Or “For example, when displayed on bacteriophage VLPs, L2 peptides can be highly immunogenic.” What, specifically did these studies show?

Response:

We agree with the reviewer that we need to clarify the focus of this review and better align the focus with the title. To further address the reviewer’s specific concerns, we have eliminated the section on the gut microbiome. The sentences given as an example have been reworded or discarded in order to better portray the information. It now reads as “There is evidence suggesting that using a platform with multivalent display allows the immunogenicity of the L2 peptide to increase, such as being displayed on a bacteriophage VLP [144,146,148]. The ability to broaden coverage with the L2 protein could potentially give more protection against HPV to populations who are more at risk, such as Native Americans.”

Comment 2. The authors state that HPV infection rates are higher in Native American populations even though vaccination rates in this population are higher than for the non-Hispanic white population. The authors conclude at the end of section 1 that they are going to discuss the accessibility of HPV vaccines in the Native American population, and later state that the higher rates of cancer may be due to “insufficient cervical cancer screenings and routine well woman exams”. More explanation is needed – if women are being vaccinated at a higher rate but not being seen by their healthcare providers, that seems contradictory. Also, a summary of data to support these claims would strengthen the manuscript.

Response: We agree with the reviewer that these statements might seem contradictory to the reader. More explanation has been added in order to allow the reader to fully comprehend the issue. One reason Native American women exhibit high rates of HPV infection and cervical cancer is because the genotypes of HPV that Native communities are infected with are not included within the current HPV vaccines. For example, HPV-51 was the most prevalent in two separate studies. However, these are only 2 communities of over 570 federally recognized tribes. As mentioned in the “’Old’ Controversy – Disproportional Impact of HPV on Native American Women” section, this gap in vaccine protection leaves Native American women more vulnerable to infection and results in a disproportionate rate of HPV infections despite vaccination compliance. This is a result of Native American populations not being considered included in studies and clinical trials for HPV vaccine efficacy. Furthermore, high vaccine compliance is only seen in younger populations because older women were not eligible. Last, HPV vaccination is performed primarily in adolescent girls and is not at the time of a well woman exam or cervical cancer screening. Therefore, that helps to explain the disparity between high vaccination rates and insufficient well woman exams and cervical cancer screening. In addition, unfortunately many women believe that HPV vaccination precludes well woman exams and cervical cancer screening.

Examples of included text are “The reasons for the high prevalence are uncertain however, it is important to note older women were not eligible for the vaccines at the time.” And “Clinical trials for HPV vaccines did not include Native populations and this bias resulted in a gap of protection. Therefore, Native American women are not as protected against prevalent HPV types as their white, non-Hispanic counterparts. As such, despite having higher vaccination rated compared to white, non-Hispanic women, there is evidence to suggest the HPV vaccines are not as effective within the Native American populations therefore resulting in a disproportionate rate of HPV infections [10].

Comment 3. Figure 1 is referred to in several places, even as support for statements that are not included in the figure. Where in Figure 1 is the information to support the sentence: “Even fewer studies mention the impact on structural racism and oppression of minoritized groups and how that may impact microbial composition [Figure 1].” Where in Figure 1 is the information to support the sentence: “Further need for studies including vaccination status, hormone status, immune markers, and microbiome composition is needed to get a better picture of the underlying mechanism for adequate vaccine efficacy [Figure 1].” 

Response: We agree with the reviewer that referencing Figure 1 was not appropriate in all circumstances and removed it as suggested. However, we edited the section “Current Strategies – Cervical Cancer Screening and HPV testing” to reference risk factors related to figure 1. It now states ”Specifically, older Native Americans have higher incidents of cervical cancer mortality. Though the reasons behind the disparities are not certain, it is likely due to various risk factors highlighted in Figure 1 [10,15,16]. The lack of screening can be due to a number of societal and environmental factors such as access to care, travel/transportation issues on the reservations and/or lack of knowledge [10,15,21].”

Comment 4. The section “New Strategy – Elucidating the Vaginal Microbiome…” doesn’t discuss any new strategies that I could determine.

Response: We have removed “new strategy” from the heading title to better reflect the focus of this section and revised it to state “Elucidating the Vaginal Microbiome in Native American Populations in the Context of HPV Infection”. This section now focuses on the need for further investigation of the vaginal microbiome in Native American populations as well as in the context of HPV infection, cervical carcinogenesis, and HPV vaccines.

Comment 5. The statement “Therefore, novel solutions have been proposed beyond antibiotic usage to help maintain homeostasis at this [cervicovaginal environment] site” seems odd. Antibiotic usage has been shown to disrupt normal flora, not maintain homeostasis. One “novel” solution discussed in this review has been around for decades, and then the authors state that these have “proven antiviral activity” but do not discuss any of the studies supporting this claim. They also do not review the status of using novel metabolites or peptides produced by lactobacilli, which would be interesting to the reader.

Response: We understand the reviewer's confusion, we have removed words such as novel to not undermine the message of preestablished therapies. We agree the message of antibiotic usage was unclear, often antibiotics can be used to treat BV in hopes that the vaginal microbiome will go back to a balanced state. However, over years of research antibiotics alone are not suitable for maintaining homeostasis post-infection or dysbiosis. This message however detracts from our statements and thus has been removed for clarity. We do mention bacterial produced metabolites for therapy, but do not mention specific metabolites we have taken the interest into account and adjusted the text.

Comment 6. In the section on “HPV, Cervical Cancer, and the Cervicovaginal Microbiome”, the authors write “The reason for the shift in dominant bacteria species may be due to antiviral immune responses [23,37].” What are these? This would be interesting to read. The next paragraph shifts focus away from HPV and cervical cancer to discuss GI bacteria and gastric cancer, which is outside the scope of this paper. A casual mention of pathogens linked to cancer is OK and should be in an introductory paragraph, not within a section devoted to HPV and the cervicovaginal microbiome. A quick search of the literature on the cervicovaginal microbiome revealed interesting papers not included that could have been added and strengthened this section – a few notable examples: Usyk et al. 2020, Curty et al. 2020, Dai et al. 2021

Response: We have taken into account the reviewer’s feedback and have modified the section to focus more in depth on the literature surrounding cervicovaginal microbiome and HPV for clearer understanding. We also added key meta-analyses as well as the ones provided by the reviewer.

Comment 7. The section on “Vaccine development and modulation of the microbiome” focuses on the gut microbiome. How would this aid in HPV vaccine development? The authors discuss “vaccines for treatment of rotavirus infection” (should be prevention, not treatment!) and discuss the link between that vaccine and the gut microbiome, but this vaccine is oral – what is the connection with the HPV vaccine, which is IM? How does the cervical microbiome affect HPV vaccination efficacy? This is not discussed. Additionally, the authors point out “increased intestinal immunity” or “increased immune responses” but don’t summarize what they are. 

Response: We thank the reviewer for their feedback and have removed the gut microbiome and discussion of GI vaccines from this section. In addition, we revised this section to focus only on the HPV vaccine.

Comment 8. In the section “Current Vaccine Targets: L1”, the authors provide a review of studies using the L2 capsid - should the section title be changed/modified to include L2? More importantly, this section ends with the statement that targeting the L2 capsid protein may be the basis of next-generation vaccination. However, the studies cited in this review are 5-10 years old (the authors don’t include newer studies, such as Yadav et al. 2020) nor discuss why these vaccines haven’t been developed. Is there a reason why this information is a decade old yet L2 capsid proteins have not been incorporated into a vaccine? What efficacy has been reported by these studies? More discussion here would strengthen this manuscript.

Response: We appreciate the author’s comment and suggestions for strengthening this section. As such, another section has been added to separate discussions of L1 and L2 for enhanced clarity. The reviewer has suggested a key paper by Yadav et al. 2020 and it has been included in this section. Additional literature has been added in order to explain the current state of L2 vaccines such as Olczak and Roden 2020. Since vaccines usually require several years to undergo the preclinical and clinical pipelines many of the studies with L2 are still in the pre-clinical setting with murine models, including the work described in Yadav et al. 2020. However, there is limited literature in regards to completed clinical trials of L2 HPV vaccinations. Table 2 in Olczak and Roden 2020 show that most vaccine trials are in preparation or in phase I clinical trial

Comment 9. The authors make a point (in a couple of places) about the cold chain requirement as a factor affecting vaccination rates. Earlier they state that vaccination rates are higher among Native Americans, which seems contradictory to their argument. But in addition, a study published by Shank-Retzlaff et al. (2006) evaluated the thermostability of Gardasil and their data show prolonged stability if kept at least at room temperature. This should be discussed.

Response: See comment above related to HPV vaccination rates in Native Americans. We have also included information on CDC and other HPV vaccine campaigns that have provided the needed infrastructure for increasing these vaccination rates that allow for appropriate storage and cold chain requirements not available on the reservation. We have mentioned the paper presented, however this may not be ideal and is included in our discussion.

Comment 10. The conclusion makes statements that have not been supported or adequately addressed throughout the review. For example: “HPV-mediated cervical cancer disproportionally impacts Native American communities due to a network of complex factors such as immunological, behavioral, social, clinical, and environmental.” Where’s the information on social factors influencing rates of cervical cancer? The authors state that one hurdle to address is access to HPV vaccines in the Native American population, yet they state that vaccination rates are higher in this population compared to the white population (and why just this comparison??). 

Response: We agree with the reviewer and expanded on risk factors throughout the manuscript. See responses to comments 2 and 3.

Minor concerns/edits:

Comment 1. In the abstract, the authors state that cervical cancer is the fourth most common type of cancer. I believe the authors meant to include “in women”. While the WHO makes this statement according to 2018 data, the CDC lists “corpus and uterus” cancer as the fourth leading cause. This should be clarified.

Response: We thank the reviewer for catching this statement. We have included “in women” for accuracy. While the CDC website does in fact list “corpus and uterus” cancer as the fourth most common type of cancer, these numbers are solely within the United States. Our statement was relating to world-wide statistics, which is why we reference WHO. This has been clarified by adding “world-wide” to our statement.

Comment 2. To be more precise, in the introduction (and other places – without line numbers it’s difficult to point out all instances) instead of saying “The main cause of cervical cancer is human papillomavirus,…” edit to read “The main cause of cervical cancer is infection with human papillomavirus,…”

Response: We thank the reviewer for making this suggestion. All phrases found written as “the main cause of cervical cancer is human papillomavirus” has been changed to read similarly to “the main cause of cervical cancer is infection with human papillomavirus”.

Comment 3. In several places, the authors write “the HPV vaccine” when there are at least four to my knowledge. Edit to using “HPV vaccines” or “HPV vaccination” (e.g., top of page 2: “Once HPV vaccination became available…” and “Despite having effective HPV vaccines available…”). Also, Cervarix is misspelled (p. 7).

Response: We thank the reviewer for pointing this out and have clarified throughout as suggested. All references to “HPV vaccine” has been changed to “HPV vaccines” to accurately refer to the multiple available vaccines. Additionally, our misspelling of Cervarix has been corrected.

Comment 4. Section 3, third line: change “species” to “genus”: “…often dominated by one particular bacterial genus, Lactobacillus,…”. Alternatively, authors could write “…often dominated by one or more species within the genus Lactobacillus…"

Response: We have revised based on the reviewer’s comment.

Comment 5. Figure 2: The legend mentions KFE8 beta-sheet fibrils but nowhere in the manuscript is a description of what KFE8 means.

Response: We thank the reviewer for catching this oversight. KFE8 refers to a specific self-assembling peptide in the original draft of the manuscript. However, the name is not necessary to include and it has been replaced with simply “a self-assembling peptide”.

Comment 6. The authors should carefully proofread the manuscript prior to resubmitting. There are fragment sentences, grammatically incorrect sentences, and tense disagreements in several places. Without line numbers, it’s hard to point them all out, but here are a few examples:

  • “Although there has been links to race and/or ethnicity having an impact on vaginal microbiome composition [41,78-80].”
  • “Such as HPV-51, the most prevalent HPV type found in two geographically separate Native American communities [10].”
  • “Deaths from cervical cancer rates dropped significantly however more needs to be done to increase vaccine uptake [1,2,7,8].” (Deaths from a rate??)
  • “It is important to note that evidence has been provided for direct and indirect effects of microbial composition importance with regards to cervical cancer development.”
  • “…some serotypes of the virus are now established as a drivers…”
  • “…mutations in BRCA1 gene…” (needs a “the” before BRCA1)

Response: We have taken the time to proofread and correct all the grammatically incorrect sentences and ensure there are no more tense disagreements and fragments in the revised document. All the examples the reviewer has given us have been fixed to ensure clarity and optimal reading comprehension.

Round 2

Reviewer 2 Report

The authors did address my comments and made significant improvements to the manuscript in terms of organization and content. I still have two major comments that either were not addressed, not fully addressed, or arose due to addition of new content in the revised manuscript:

1.      I still find that the argument of lack of a cold-chain on reservations affecting vaccination is not fully supported without referencing the study that has evaluated efficacy of HPV vaccines at room temperature out to 130 months. The studies that these authors cite to support requirements for refrigeration are not for HPV or VLPs (ref 154 does not reference HPV or VLPs and refs 155-156 focus on adenoviral vectored vaccines, which do require cold, but are quite different in composition to HPV). I feel that this section is inaccurate and misleading as written. While refrigeration is probably best, the authors need to cite/include published data on the requirement for cold with HPV and/or VLP vaccines if the study I provided in the first review is contradictory to other studies or doesn't provide enough evidence to support their claims.

 2.    I appreciated the updated L1 and L2 sections - much better.  One question remains for me, however. You make this statement on page 13: “The ability to broaden coverage with the L2 protein could potentially give more protection against HPV to populations who are more at risk, such as Native Americans.” The L2 consensus sequence neutralized five HPV strains (16, 18, 31, 45, and 58) but not HPV-51.  However, HPV-51 is the strain you state is most prevalent in Native American women.  So, does using an L2-based vaccine broaden protection, since these five strains neutralized are in the Gardasil-9 vaccine?  How would using an L2 vaccine help the Native American population differently than existing vaccines?

3.     I appreciate the new Section 3, but puzzled at this addition (page 7): Usyk et al., observed that Gardnerella sp and diverse microbiota were more frequently observed in patients that progressed to cervical intraepithelial neoplasia (CIN) [53]. It is thought that a dysbiotic vaginal microbiome may contribute to this progression of CIN by modulating the host immune response, causing DNA damage or directly disrupting the cell barrier for oncoviruses like HPV to infect the host more readily [54].  Are the authors saying that a diverse microbiota is equivalent to a dysbiotic vaginal microbiome?  Or is the contribution of Gardnerella to the overall microbiota and/or non-Lactobacillus dominated or L. iners dominated microbiome profiles what is contributing to CIN?  It just seems vague/unclear to me as written but I do like this addition.  

While the authors did fix some of the grammatical errors from the previous version, there are still several typos and/or grammatically incorrect sentences in the manuscript, including in the new text inserted.  I’ve tried to list out several so that the authors can see the extent of what I see, but I am sure I didn’t catch all of them.  I’d like to suggest that the authors submit for proofreading either through the journal or contract with someone that can read for grammar and not science to help facilitate publishing a manuscript that is well written.    

1.       Throughout manuscript: make sure all binomials (Genus, species) names of bacteria are italicized.

2.       Page 3, fourth line from bottom: Replace “rated” with “rates”

3.       Page 6, top:  “In fact, patients with persistent high risk HPV infections had higher prevalence of co-occurring condition bacterial vaginosis (BV) compared to patients with cleared HPV infections [32,43].”  Should read “In fact, patients with persistent high-risk HPV infections had a higher prevalence of concurrent bacterial vaginosis (BV) compared to patients with cleared HPV infections.”  Or is this still not entirely correct?

4.       Page 6, first full paragraph: This sentence reads awkward:  “Brusselaers et al., performed a systematic review of 14 observational studies reporting on incident HPV, persistence and/or related cervical disease in women with or without dysbiotic cervicovaginal microbiome.”  Seems awkwardly worded.

5.       Page 6, middle of first paragraph. This sentence seems confusing, grammatically incorrect (and a run-on sentence), and needs italicization of bacterial names:  “In this report of 11 articles that fit the criteria described in the article, they found once again that dysbiotic vaginal microbiome and abundance of Lactobacillus iners had increased risk of infection of HPV and/or progression to cervical cancer compared to vaginal microbiome profiles that contained Lactobacillus crispatus the studies are foundational [43], but have limitations in regards to original studies with regard to confounding factors of cohorts such as younger age women, women with more sexual partners, and high-risk sexual behavior.” 

a.       “In this report of 11 articles that fit the criteria described in the article, they…” is not very clear.

b.       Who is “they” in this sentence (I assume it is reference 43) but next sentence reads “…not mentioned by the author..” (single author)?

6.       Page 6, second full paragraph: First sentence is grammatically incorrect: “Supporting these findings that HPV clearance is delayed when patients were diagnosed with BV, dysbiotic vaginal microbial condition [28,46]”. The second sentence uses “BVAB” but this was deleted from the paragraph at the top of page 6.  Maybe state “..and a dominance of bacterial vaginosis-associated bacteria (BVAB), such as…”  The acronym BVAB was used again in other places throughout the manuscript.  Need to spell out at first use.

7.       Page 6: “…from NK and epithelial cells production…”  should be “cell”.  Can you expand upon this statement too?  How does antimicrobial peptide production affect viruses, and how would that shift the flora?

8.       Page 8:  “Although there has been links…”  Replace “has” with “have”

9.       Page 8:  “Overall, in order to better HPV and cervical cancer health disparities…”  The authors struck out “resolve the”.  Add back in?

10.   Page 8: “…a low-cost strategy with minimal known side effects and [104-108].”  Delete “and”

11.   Page 9: “Some literature has investigated whether there is a link between specific microbes from gut microbiome samples could be predictive of vaccine efficacy [125-127].”  Missing something here.

12.   Page 9: “…were less likely to illicit…”  Replace “illicit” with “elicit”. 

13.   Page 9-10: “Hormonal status has shown to be a vital factor in the immune response to mucosal pathogens, likewise the day at which vaccination is given and day during menstrual cycle has been shown to contribute to HPV vaccination efficacy and mounting a sufficient immune response that offers long lasting protection [131,132].”  Run-on sentence and unclear.

14.   Page 11: “While there has been some program initiative to lower the cost for the lowest income countries, preventative measures and treatment is still limited…”  Edit for tenses (initiatives, treatment/treatments are)

15.   Bottom, Page 13: “While targeting the HPV L2 protein with a consensus sequence could be the basis of next-generation HPV vaccines for broader protection, more studies are required As of 2020 there are several L2 vaccines in the clinical trial stage but most are in phase I [153].”  Put in a semicolon and lowercase “as” as a suggestion?

Author Response

We have revised the manuscript to address the concerns. We thank the reviewer and believe this work has strengthened and clarified our manuscript. Below we provide details about how we responded to the suggestions and comments.

We appreciate your consideration of our manuscript.

  1. I still find that the argument of lack of a cold chain on reservations affecting vaccination is not fully supported without referencing the study that has evaluated efficacy of HPV vaccines at room temperature out to 130 months. The studies that these authors cite to support requirements for refrigeration are not for HPV or VLPs (ref 154 does not reference HPV or VLPs and refs 155-156 focus on adenoviral vectored vaccines, which do require cold, but are quite different in composition to HPV). I feel that this section is inaccurate and misleading as written. While refrigeration is probably best, the authors need to cite/include published data on the requirement for cold with HPV and/or VLP vaccines if the study I provided in the first review is contradictory to other studies or doesn't provide enough evidence to support their claims.

Response:

We included additional references that discuss both the thermostability of Gardasil and VLPs. As such, we added the statement, “The thermostability of Gardasil 4® was evaluated at varying temperatures (25, 37, and 42 oC), of which, the half-life decreased from 130 months to approximately three months [155]. When spray dried, Gardasil 9® maintained protection in mice indicating the vaccine may be stored up to 42 oC for up to 3 months without losing efficacy [159].”  

We also expanded on the thermostability of self-assembling peptides and their potential as vaccines by adding the statement “Peptides are promising vaccine platforms due to their thermostability and ability to be stored for extended periods of time when lyophilized, eliminating the cold chain need [155]. Previous studies that heated lyophilized self-assembling peptides at 45 °C for one to five weeks were stable and showed indistinguishable morphology. Likewise, the immunogenicity in mouse models did not diminish, even after heating the peptide to 45 °C for six months [155]. In addition, self-assembling peptides can also withstand varying solvents and pH unlike current HPV vaccines [155,165,166].”

Below are 3 references added in the above statements that studied the thermostability of VLPs.

Ref 153. Shank-Retzlaff et al 10.4161/hv.2.4.2989

“The half-life of the vaccine is estimated to be 130 months or longer at temperatures up to 25 degrees C. At 37 degrees C, the half-life is predicted to be 18 months and at 42 degrees C, the half-life is predicted to be approximately three months.”

Ref 154. Frietze et al 10.1016/j.coviro.2016.03.001

“Although VLPs are generally highly stable, VLP types that can withstand variations in temperature and other conditions will likely be advantageous both during manufacturing and distribution. Engineering VLPs to withstand storage or transport at diverse temperatures can facilitate the logistics of vaccine deployment, particularly in areas of the developing world where it is difficult to maintain a cold chain. Additionally, improving the stability of VLPs reduces the cost of vaccines by allowing a longer shelf-life and also reduces energy costs associated with cold-storage.”

Ref 157. Kunda et al doi.org/10.1080/21645515.2019.1593727

“The immunogenicity studies were performed using Gardasil® 9 as a whole antigen, and not individual HPV types, for ELISA. At the dose tested, the spray dried vaccine conferred protection against HPV following storage at temperatures up to 40°C. In addition to the spray-dried vaccine, our studies revealed that the Gardasil® 9 vaccine, as currently marketed, may be stored and transported at elevated temperatures for up to 3 months without losing efficacy, especially against HPV16.”

  1. I appreciated the updated L1 and L2 sections - much better. One question remains for me, however. You make this statement on page 13: “The ability to broaden coverage with the L2 protein could potentially give more protection against HPV to populations who are more at risk, such as Native Americans.” The L2 consensus sequence neutralized five HPV strains (16, 18, 31, 45, and 58) but not HPV-51.  However, HPV-51 is the strain you state is most prevalent in Native American women.  So, does using an L2-based vaccine broaden protection, since these five strains neutralized are in the Gardasil-9 vaccine?  How would using an L2 vaccine help the Native American population differently than existing vaccines?

Response:

We agree with the reviewer that clarification is needed on the L2 consensus sequence and coverage of genotypes. Thus, we added the following statements, “Experiments showed that mice immunized with this consensus L2 sequence (GTGGRTGYVPLGTRPPTVVDV) on VLPs were able to create antibodies and neutralize a wide spectrum of HPVs, specifically HPV-5, 6, 16, 18, 31, 33, 35, 39, 45, 51, 53, and 58 [142,151].”

“In another study, the authors showed that despite inducing a lower antibody titer count than HPV-16 or HPV-18 L2 proteins, the antibodies from the L2 consensus sequence were able to neutralize HPV-16, 18, 31, 45, and 58 [144].”

“Current studies are underway that aim to broaden immunity and increase thermostability by using synthetic self-assembling peptides and the previously reported HPV consensus sequence [147].”

References included in the statements above that demonstrate HPV-51 protection in the L2 consensus are highlighted below.

Ref 142. Tumban et al 10.1371/journal.pone.0049751

Table 1. Shows a 2.2-fold reduction of HPV-51 using PP7 VLPs

Ref 151. Tyler et al 10.1586/14760584.2014.865523

“Using the PsV in vivo challenge model, we observed significant protection against HPV types 5, 6, 16, 18, 31, 33, 35, 39, 45, 51, 53, and 58 in mice vaccinated with our L2 displaying VLPs – refers to previous citation

Ref. 147 Yadav et al 10.3390/v13061113.

“We developed mixed MS2-L2 VLPs (MS2-31L2/16L2 VLPs and MS2-consL2 (69-86) VLPs) in a previous study.

  1. I appreciate the new Section 3, but puzzled at this addition (page 7): Usyk et al., observed that Gardnerella sp and diverse microbiota were more frequently observed in patients that progressed to cervical intraepithelial neoplasia (CIN) [53]. It is thought that a dysbiotic vaginal microbiome may contribute to this progression of CIN by modulating the host immune response, causing DNA damage or directly disrupting the cell barrier for oncoviruses like HPV to infect the host more readily [54]. Are the authors saying that a diverse microbiota is equivalent to a dysbiotic vaginal microbiome?  Or is the contribution of Gardnerella to the overall microbiota and/or non-Lactobacillus dominated or L. iners dominated microbiome profiles what is contributing to CIN?  It just seems vague/unclear to me as written, but I do like this addition. 

Response:

We appreciate the reviewer’s comment and have revised the statement for clarity:

“Usyk et al., observed that dysbiotic microbiota and Gardnerella spp, key players in BV, were more frequently observed in patients that progressed to cervical intraepithelial neoplasia (CIN) [51]”

While the authors did fix some of the grammatical errors from the previous version, there are still several typos and/or grammatically incorrect sentences in the manuscript, including in the new text inserted.  I’ve tried to list out several so that the authors can see the extent of what I see, but I am sure I didn’t catch all of them.  I’d like to suggest that the authors submit for proofreading either through the journal or contract with someone that can read for grammar and not science to help facilitate publishing a manuscript that is well written.   

Minor comments addressed below. The updates are included for each comment.

  1. Throughout manuscript: make sure all binomials (Genus, species) names of bacteria are italicized.
    1. Addressed throughout the manuscript

  1. Page 3, fourth line from bottom: Replace “rated” with “rates”
    1. As such, despite having higher vaccination rates compared to white, non-Hispanic women, there is evidence to suggest the HPV vaccines are not as effective within the Native American populations therefore resulting in a disproportionate rate of HPV infections [10].

  1. Page 6, top:  “In fact, patients with persistent high-risk HPV infections had higher prevalence of co-occurring condition bacterial vaginosis (BV) compared to patients with cleared HPV infections [32,43].”  Should read “In fact, patients with persistent high-risk HPV infections had a higher prevalence of concurrent bacterial vaginosis (BV) compared to patients with cleared HPV infections.”  Or is this still not entirely correct?
    1. In fact, patients with persistent high risk HPV infections had higher prevalence of bacterial vaginosis (BV) compared to patients that cleared HPV infections [32,39].

  1. Page 6, first full paragraph: This sentence reads awkward:  “Brusselaers et al., performed a systematic review of 14 observational studies reporting on incident HPV, persistence and/or related cervical disease in women with or without dysbiotic cervicovaginal microbiome.”  Seems awkwardly worded.
    1. Brusselaers et al., performed a systematic review of 14 observational studies reporting on incident HPV, HPV persistence, and cervical disease in women with or without a dysbiotic cervicovaginal microbiome.

  1. Page 6, middle of first paragraph. This sentence seems confusing, grammatically incorrect (and a run-on sentence), and needs italicization of bacterial names:  “In this report of 11 articles that fit the criteria described in the article, they found once again that dysbiotic vaginal microbiome and abundance of Lactobacillus iners had increased risk of infection of HPV and/or progression to cervical cancer compared to vaginal microbiome profiles that contained Lactobacillus crispatus the studies are foundational [43], but have limitations in regards to original studies with regard to confounding factors of cohorts such as younger age women, women with more sexual partners, and high-risk sexual behavior.”
    1. In this report of 11 articles, the authors that cervicovaginal microbiome profiles containing dysbiotic microbes or Lactobacillus iners had increased risk of HPV infection and/or progression to cervical cancer compared with  cervicovaginal microbiome profiles that  were dominated by Lactobacillus crispatus [39]. These studies are foundational but have limitations with regard to confounding factors within the cohort such as including younger age women, women with more sexual partners, and high-risk sexual behavior.
  2. “In this report of 11 articles that fit the criteria described in the article, they…” is not very clear.
    1. See response to #5

  1. Who is “they” in this sentence (I assume it is reference 43) but next sentence reads “…not mentioned by the author..” (single author)?
    1. See response to #5

  1. Page 6, second full paragraph: First sentence is grammatically incorrect: “Supporting these findings that HPV clearance is delayed when patients were diagnosed with BV, dysbiotic vaginal microbial condition [28,46]”. The second sentence uses “BVAB” but this was deleted from the paragraph at the top of page 6.  Maybe state “..and a dominance of bacterial vaginosis-associated bacteria (BVAB), such as…”  The acronym BVAB was used again in other places throughout the manuscript.  Need to spell out at first use.
    1. Other reports also found that HPV clearance is delayed when patients were diagnosed with BV (bacterial vaginosis), a dysbiotic vaginal microbial condition [28,43] Roughly 43% of women with persistent HPV infection had a decrease in lactobacilli and a dominance of BV-associated bacteria (BVAB) such as Gardnerella, Prevotella, Atopobium (reclassified as Fannyhessea), and Megasphaera species compared to 7.4% of women with cleared HPV infection [32].

  1. Page 6: “…from NK and epithelial cells production…”  should be “cell”.  Can you expand upon this statement too?  How does antimicrobial peptide production affect viruses, and how would that shift the flora?
    1. The shift in dominant bacterial species may be due to antiviral immune responses from NK and epithelial cells that are able to produce antimicrobial peptides (AMPs) [44,45]. Although AMPs are mostly associated with bacterial inhibition, new studies suggest an ability to inhibit viral pathogens as well [49].

  1. Page 8:  “Although there has been links…”  Replace “has” with “have”
    1. Although there have been links to race and/or ethnicity being associated with specific vaginal microbiome composition [50,93-95],

  1. Page 8:  “Overall, in order to better HPV and cervical cancer health disparities…”  The authors struck out “resolve the”.  Add back in?
    1. Overall, in order to better resolve the HPV and cervical cancer health disparities amongst Native American populations further vaginal microbiome studies are required across tribal communities, as well as incorporation of additional social, behavioral and societal factors that may impact these minority groups.

  1. Page 8: “…a low-cost strategy with minimal known side effects and [104-108].”  Delete “and”
    1. One approach to re-establishing a lactobacilli dominant vaginal environment is through the delivery of probiotics and/or prebiotics as a low-cost strategy with minimal known side effects [103-107].

  1. Page 9: “Some literature has investigated whether there is a link between specific microbes from gut microbiome samples could be predictive of vaccine efficacy [125-127].”  Missing something here.
    1. Some literature has investigated whether the composition of the gut microbiome could be predictive of vaccine efficacy [124-126].

  1. Page 9: “…were less likely to illicit…”  Replace “illicit” with “elicit”.
    1. A recent study by Ravilla et al. revealed that cervicovaginal microbiome profiles with high abundance of Prevotella, Caldithrix, and Nitrospira were less likely to elicit a protective immune response post-HPV vaccination, however sample size and bacterial classification methods were limitations to the study cohort [127].

  1. Page 9-10: “Hormonal status has shown to be a vital factor in the immune response to mucosal pathogens, likewise the day at which vaccination is given and day during menstrual cycle has been shown to contribute to HPV vaccination efficacy and mounting a sufficient immune response that offers long lasting protection [131,132].”  Run-on sentence and unclear.
    1. Hormonal status can impact the immune response to mucosal pathogens [120]. For example, the day at which vaccination is given and day during the menstrual cycle has been shown to contribute to HPV vaccination efficacy and mounting a sufficient immune response that offers long lasting protection [130,131].

  1. Page 11: “While there has been some program initiative to lower the cost for the lowest income countries, preventative measures and treatment is still limited…”  Edit for tenses (initiatives, treatment/treatments are)
    1. While there have been program initiatives to lower the cost for low-income countries, preventative measures and treatment are still limited, resulting in higher cervical cancer burden in countries such as Malawi, Uganda, the United Republic of Tanzania, Zimbabwe, and Zambia [1,139-141]

  1. Bottom, Page 13: “While targeting the HPV L2 protein with a consensus sequence could be the basis of next-generation HPV vaccines for broader protection, more studies are required As of 2020 there are several L2 vaccines in the clinical trial stage but most are in phase I [153].”  Put in a semicolon and lowercase “as” as a suggestion?
    1. While targeting the HPV L2 protein with a consensus sequence could be the basis of next-generation HPV vaccines for broader protection [147], more studies are required; as of 2020, there are several L2 vaccines in the clinical trial stage but most are in Phase I [152].

Response:

We thoroughly read the manuscript and ran through a writing assistant to check for more errors. The most significant edits are listed below.

Page 5. Cervicovaginal profiles with high abundance of lactobacilli can prevent HPV infection by increasing acidity levels via production of lactic acid [32]. A cervicovaginal microbiome that is more diverse with lower abundance of lactobacilli can increase the risk of HPV acquisition, persistent infection, and progression of neoplasia. Neoplasia may also alter the microbiome further, which could lead to clinical sequelae [28,39,40]. 

Page 5. In addition, it was not mentioned by the authors that the cohort demographics included predominantly non-Hispanic, white women from the United States and European countries.

We removed the following statements due to redundancy,

Page 6. “A decrease in lactobacilli and an increase in other anaerobes such as Streptococcus, Gardnerella, Megasphaera, and Anaerococcus have also been associated with cervical dysplasia [46,63].”

Page 6. Lactobacilli dominance has also been observed to increase cervical intraepithelial neoplasia regression, while profiles with Megaspheara, Prevotella, Gardnerella, and BVAB were less likely [78].

Page 6. The beginning of section “Elucidating the Cervicovaginal…” was modified to now read as, “The cervicovaginal microbiome can be impacted by many external factors such as hormonal changes as in pregnancy and menopause, behavioral practices such as smoking and douching, and external/xenobiotic factors such as antibiotic usage [40,86-88]. Other factors such as host genetics can also play a role in microbial composition and in a recent study, patients with mutations in the BRCA1 gene associated with vaginal profiles with less than 50% abundance of Lactobacillus species [81,89-92]. Although there have been links to race and/or ethnicity being associated with specific vaginal microbiome composition [50,93-95], it is not fully understood whether these are biologically relevant features or features that relate to socioeconomic status, hygiene practices, geographic location, and diet [96]. Even fewer studies mention the impact on structural racism and oppression of minoritized groups and how that may impact microbial composition [97]. Further studies that engaged Native Americans and Indigenous populations often over generalize these groups and do not account for distinct cultural and geographic differences of these racial/ethnic groups [98]. Thus, it is important to not draw conclusions from one cohort or tribal community and generalize to all Native American communities.”

Page 8. Added the statements, “Despite the findings from cervicovaginal microbiome and HPV vaccines, Native American women were not included in these studies.

Cervicovaginal vaccine development for the general population, and diverse communities such as Native Americans, it is important to note the complex interplay of systemic and local mucosal immunological factors, cervicovaginal microbiome composition and hormonal status (age, treatment, or contraceptive usage) on vaccine efficacy [129].”

Page 12. Add the statement, “Within healthcare facilities, it is common practice to discard vaccines that have been exposed to high temperatures to ensure patient health.”

Page 12. Restructured the paragraph to state, “Amphipathic peptides that have a sequence of alternating hydrophilic and hydrophobic residues spontaneously assemble into β-sheet bilayers when in aqueous solution seen in Figure 2 [158,162,163]. This assembly is driven by the hydrophobic residues burying the side chains inside the bilayer as hydrophilic residues form the outer layer [161,162]. A variety of structures can be formed from self-assembling β-sheets such as tapes, ribbons, and fibrils depending on the density of sheets that pack together [161,163]. Amphipathic β-sheets alone have many purposes in drug delivery such as hydrogel scaffolds and nanofibers [161,162,164].“